# Memorization and Optimization in Deep Neural Networks with Minimum Over-parameterization

Simone Bombari[*], Mohammad Hossein Amani[†], Marco Mondelli[*]

## Abstract

The Neural Tangent Kernel (NTK) has emerged as a powerful tool to provide memorization, optimization and generalization guarantees in deep neural networks. A line of work has studied the NTK spectrum for two-layer and deep networks with at least a layer with $\Omega(N)$ neurons, $N$ being the number of training samples. Furthermore, there is increasing evidence suggesting that deep networks with sub-linear layer widths are powerful memorizers and optimizers, as long as the number of parameters exceeds the number of samples. Thus, a natural open question is whether the NTK is well conditioned in such a challenging sub-linear setup. In this paper, we answer this question in the affirmative. Our key technical contribution is a lower bound on the smallest NTK eigenvalue for deep networks with the *minimum possible over-parameterization*: up to logarithmic factors, the number of parameters is $\Omega(N)$ and, hence, the number of neurons is as little as $\Omega(\sqrt{N})$. To showcase the applicability of our NTK bounds, we provide two results concerning memorization capacity and optimization guarantees for gradient descent training.

## 1 Introduction

Training a neural network is a non-convex problem that exhibits disconnected local minima [8, 57, 70]. Yet, in practice gradient descent (GD) and its variants routinely find solutions with zero training loss [72]. A framework to understand this phenomenon comes from the Neural Tangent Kernel (NTK). This quantity was introduced in [33], where it was proved that, during GD training, the network follows the kernel gradient of the functional cost with respect to the NTK. Furthermore, as the layer widths go large, the NTK converges to a deterministic limit which stays constant during training. Hence, in this infinite-width limit, it suffices that the smallest eigenvalue of the NTK is bounded away from 0 for gradient descent to reach zero loss. Going to finite widths, a recipe to prove GD convergence can be summarized as follows: show that *(i)* the NTK is well conditioned at initialization, and *(ii)* the NTK has not changed significantly by the time GD has reached zero loss (see e.g. [49, 17, 11]). A number of papers have exploited this recipe for networks with progressively smaller over-parameterization: two-layer networks [22, 50, 63, 69, 62], deep networks with polynomially wide layers [5, 21, 73, 74], and deep networks with a single wide layer [47, 43]. Besides optimization, showing that the NTK is well conditioned directly implies a result on memorization capacity [42, 48], and the smallest eigenvalue of the NTK has also been related to generalization [6].

In [48], it is shown that, given $N$ training samples, a single layer with $\Omega(N)$ neurons suffices for the NTK to be well conditioned in networks of arbitrary depth. However, there is increasing evidence that, in the challenging setup in which the layer widths are *sub-linear* in $N$, neural networks still memorize the training data [14, 71, 66], reach zero loss under GD training in the two-layer setting [61, 50, 42], and in the deep case, GD explores a nicely behaved region of the loss landscape [44]. This is in agreement with simple back-of-the-envelope calculations on CIFAR-10 and ImageNet: CIFAR-10 has $N = 50000$ images and roughly $10^6$ parameters suffice to fit random labels [72];

---

[*]Institute of Science and Technology Austria (ISTA). Emails: `{simone.bombari, marco.mondelli}@ist.ac.at`.

[†]EPFL, Switzerland. Email: `mh.amani1998@gmail.com`.

36th Conference on Neural Information Processing Systems (NeurIPS 2022).

furthermore, in order to fit random labels to a subset of $1.2 \cdot 10^6$ ImageNet data points, $2.4 \cdot 10^7$ parameters are enough [72]. These numbers suggest that having a number of parameters of the same order as the dataset size is much closer to practice than having a number of neurons of that order. We also note that, by counting degrees of freedom or bounding the VC dimension [10], $\Omega(N)$ parameters (and, therefore, $\Omega(\sqrt{N})$ neurons) are in general necessary to fit $N$ data points. This naturally brings forward the following open question:

*Is the NTK well conditioned for deep networks with the minimum possible over-parameterization (i.e., containing $\Omega(N)$ parameters corresponding to $\Omega(\sqrt{N})$ neurons)?*

**Main contributions.** In this paper, *we settle this open question* for a large class of deep networks. We consider *(i)* a smooth activation function, *(ii)* i.i.d. data satisfying Lipschitz concentration (e.g., data with a Gaussian distribution, uniform on the sphere/hypercube, or obtained via a Generative Adversarial Network), *(iii)* a standard initialization of the weights (e.g., He's or LeCun's initialization), and *(iv)* a loose pyramidal topology in which the layer widths can increase by at most a multiplicative constant as the network gets deep. Then, in Theorem 3.1 we show that the NTK is well conditioned under the *minimum possible over-parameterization* requirement: the number of parameters between the last two layers has to be $\tilde{\Omega}(N)$ or, equivalently, the number of neurons has to be $\tilde{\Omega}(\sqrt{N})$, where $\tilde{\Omega}$ includes extra logarithmic factors. We achieve this goal by giving a lower bound on the smallest eigenvalue of the NTK. This lower bound is tight when all the layer widths are of the same order.

Our NTK bounds open the way towards understanding the behavior of deep networks with minimum over-parameterization. In particular, an immediate consequence of the fact that the NTK is well conditioned is a result on the memorization capacity (Corollary 4.1). Furthermore, by suitably choosing the initialization, we provide convergence guarantees for gradient descent training (Theorem 4.2). Finally, we highlight that, in order to obtain our bounds on the smallest NTK eigenvalue, we give a number of tight estimates on the $\ell_2$ norms of feature vectors and of their centered counterparts, which may be of independent interest.

**Proof ideas.** To prove Theorem 3.1, we restrict to the kernel $K_{L-2} = J_{L-2}J_{L-2}^\top$, where $J_{L-2}$ is the Jacobian of the output w.r.t. the parameters between the last two hidden layers. This suffices as $K_{L-2}$ is a lower bound on the NTK in the positive semi-definite (PSD) sense. We note that the $i$-th row ($i \in \{1, \ldots, N\}$) of $J_{L-2}$ is given by the Kronecker product $\otimes$ between the feature vector at layer $L-2$ and the backpropagation term from the same layer. One key technical hurdle is to center $J_{L-2}$, so that its rows have the form $u \otimes v - \mathbb{E}[u \otimes v]$, where $\mathbb{E}[u] = \mathbb{E}[v] = 0$, and all expectations are taken with respect to the (random) training data. To do so, we perform three steps of centering: *(i)* we center the feature vectors (corresponding to $u$), *(ii)* we center the backpropagation terms (corresponding to $v$), and *(iii)* we center again the whole row (corresponding to $u \otimes v$). These centering steps are approximate in the sense that the centered matrix is *not necessarily* a lower bound (in the PSD sense) on the original one. However, we are able to control the operator norm of the difference, and show that it scales slower than the smallest eigenvalue of the centered kernel. At this point, we leverage the structure of the rows of the centered Jacobian to bound their sub-exponential norm via a version of the Hanson-Wright inequality for (weakly) correlated random vectors [1]. Finally, after providing also a tight estimate on the $\ell_2$ norms of such rows, we can exploit a result from [2] to lower bound the smallest singular value of a matrix whose rows are independent random vectors with well controlled sub-exponential and $\ell_2$ norms.

Existing work bounds the smallest NTK eigenvalue for networks with two layers [61, 42], or deep networks with a layer containing $\Omega(N)$ *neurons* [48]. In particular, [61] also exploits the results [1, 2]. However, the centering of the Jacobian is achieved via a combination of whitening and dropping rows, which appears to be difficult to generalize to deep networks. In contrast, the 3-step centering we described above applies to networks of arbitrary depth $L$. A different approach is put forward in [42], and it is based on a decomposition of the kernel via spherical harmonics. This technique allows to obtain the exact limit of the smallest NTK eigenvalue, it has been used to analyze random feature models [26, 41, 25, 40] and to obtain generalization bounds for two-layer networks (see again [42]). However, understanding how to carry out such a decomposition in the multi-layer setup is an open problem. Finally, [48] considers the deep case, and it relates the smallest eigenvalue of the NTK to the smallest singular value of a feature matrix. As feature matrices are full rank only when the number of neurons is $\Omega(N)$, this approach is inherently limited to networks with a linear-width layer.

The rest of the paper is organized as follows: Section 2 discusses the problem setup and our model assumptions; Section 3 presents our main result on the smallest NTK eigenvalue and gives a roadmap of the argument; Section 4 provides two applications of our NTK bounds: memorization capacity and gradient descent training; Section 5 discusses additional related work, and Section 6 provides some concluding remarks. The details of the proofs are deferred to the appendices.

## 2 Preliminaries

**Neural network setup.** We consider an $L$-layer neural network with feature maps $f_l : \mathbb{R}^d \to \mathbb{R}^{n_l}$ defined for every $x \in \mathbb{R}^d$ as

$$f_l(x) = \begin{cases} x, & l = 0, \\ \phi(W_l^\top f_{l-1}), & l \in [L-1], \\ W_L^\top f_{L-1}, & l = L. \end{cases} \tag{1}$$

Here, $W_l \in \mathbb{R}^{n_{l-1} \times n_l}$ is the weight matrix at layer $l$, $\phi$ is the activation function and, given an integer $n$, we use the shorthand $[n] = \{1, \ldots, n\}$. We assume that the network has a single output, *i.e.* $n_L = 1$ and $W_L \in \mathbb{R}^{n_{L-1}}$, and for consistency we have $n_0 = d$. Let $g_l : \mathbb{R}^d \to \mathbb{R}^{n_l}$ be the pre-activation feature map so that $f_l(x) = \phi(g_l(x))$ for $l \in [L]$. We define $g_0(x) = f_0(x) = x$. Let $X = [x_1, \ldots, x_N]^\top \in \mathbb{R}^{N \times d}$ be the data matrix containing $N$ samples in $\mathbb{R}^d$, $\theta = [\text{vec}(W_1), \ldots, \text{vec}(W_L)]$ be the vector of the parameters of the network, and $F_L(\theta) = [f_L(x_1), \ldots, f_L(x_N)]^\top$ be the network output. We denote by $J$ the Jacobian of $F_L$ with respect to all the parameters of the network:

$$J = \left[ \frac{\partial F_L}{\partial \text{vec}(W_1)}, \ldots, \frac{\partial F_L}{\partial \text{vec}(W_L)} \right] \in \mathbb{R}^{N \times \sum_{l=1}^{L} n_{l-1} n_l}. \tag{2}$$

Our key object of interest is the *empirical Neural Tangent Kernel (NTK) Gram matrix*, denoted by $K \in \mathbb{R}^{N \times N}$ and defined as:

$$K = JJ^T = \sum_{l=1}^{L} \left[ \frac{\partial F_L}{\partial \text{vec}(W_l)} \right] \left[ \frac{\partial F_L}{\partial \text{vec}(W_l)} \right]^\top. \tag{3}$$

In [33], it is shown that, as $n_l \to \infty$ for all $l \in [L-1]$, $K$ converges to a deterministic *limit*, which stays constant during gradient descent training. The focus of this paper is on the *finite-width* behavior of the empirical NTK (3). Quantitative bounds for the NTK convergence rate can be obtained from [7, 15]. However, these bounds lead to a significant over-parameterization requirement, see the discussion at the end of Section 3 in [48]. Here, our main result consists in showing that the NTK is well conditioned for a class of neural networks with the *minimum possible over-parameterization*, i.e., $\Omega(N)$ parameter or, equivalently, $\Omega(\sqrt{N})$ neurons.

**Weight and data distribution.** We consider the following initialization of the weight matrices: $(W_l)_{i,j} \sim_{\text{i.i.d.}} \mathcal{N}(0, \beta_l^2 / n_{l-1})$ for $l \in [L-1], i \in [n_{l-1}], j \in [n_l]$, where $\beta_l$ is a numerical constant independent of the layer widths. This covers the popular cases of He's and LeCun's initialization [27, 30, 36]. For the last layer, we assume that $(W_L)_i \sim_{\text{i.i.d.}} \mathcal{N}(0, \beta_L^2)$ for $i \in [n_{L-1}]$. Throughout the paper, we let $(x_1, \ldots, x_N)$ be $N$ i.i.d. samples from the data distribution $P_X$, which satisfies the conditions below.

**Assumption 2.1** (Data scaling). *The data distribution $P_X$ satisfies the following properties:*

1. $\int \|x\|_2 \, dP_X(x) = \Theta(\sqrt{d})$.

2. $\int \|x\|_2^2 \, dP_X(x) = \Theta(d)$.

3. $\int \left\| x - \int x' \, dP_X(x') \right\|_2^2 \, dP_X(x) = \Omega(d)$.

**Assumption 2.2** (Lipschitz concentration). *The data distribution $P_X$ satisfies the* Lipschitz con-centration property. *Namely, there exists an absolute constant $c > 0$ such that, for every Lipschitz continuous function $\varphi : \mathbb{R}^d \to \mathbb{R}$, we have $\mathbb{E}|\varphi(X)| < +\infty$, and for all $t > 0$,*

$$\mathbb{P}\left( \left| \varphi(x) - \int \varphi(x') \, dP_X(x') \right| > t \right) \leq 2e^{-ct^2 / \|\varphi\|_{\text{Lip}}^2}.$$

Assumption 2.1 is simply a scaling of the training data points and their centered counterparts. Assumption 2.2 covers a number of important cases, e.g., standard Gaussian distribution [65], uniform distributions on the sphere and on the unit (binary or continuous) hypercube [65], data produced via a Generative Adversarial Network (GAN)[3] [59], and more generally any distribution satisfying the log-Sobolev inequality with a dimension-independent constant. We also remark that Assumption 2.2 is rather common in the related literature [48], or it is even replaced by a stronger requirement (e.g., Gaussian distribution or uniform on the sphere) [42].

**Assumption 2.3** (Activation function). *The activation function $\phi$ satisfies the following properties:*

1. *$\phi$ is a non-linear (and therefore also non-constant) $M$-Lipschitz function;*

2. *its derivative $\phi'$ is a $M'$-Lipschitz function;*

These requirements are satisfied by common activations, e.g. smoothed ReLU, sigmoid, or $\tanh$.

**Assumption 2.4** (Network topology). *The network satisfies a loose pyramidal topology condition, i.e. $n_l = \mathcal{O}\left(n_{l-1}\right)$, for all $l \in [L-1]$.*

A *strict* pyramidal topology (namely, non-increasing layer widths) has been considered in prior work concerning the loss landscape [45, 46] and gradient descent training [47]. Our Assumption 2.4 requires a *loose* pyramidal topology in the sense that, as we go deep, the layer widths are allowed to increase by a constant multiplicative factor. We also note that the widths of neural networks used in practice are often large in the first layers and then start decreasing [28, 60].

**Assumption 2.5** (Over-parameterization). *We have that*

$$N \cdot \log^8 N = o(n_{L-2}n_{L-1}). \tag{4}$$

*Furthermore, there exists $\gamma > 0$ such that*

$$N^\gamma = \mathcal{O}\left(n_{L-1}\right). \tag{5}$$

Condition (4) requires the number of parameters between the last two hidden layers to be linear in the number of samples, up to logarithmic factors. This represents our *key over-parameterization condition*. When all the widths have the same scaling, (4) reduces to $N = \tilde{o}(d^2)$. This is satisfied by several "standard" datasets, such as MNIST ($N = 60000$, $d = 784$), CIFAR-10 ($N = 5 \cdot 10^4$, $d = 3 \cdot 32^2$), and ImageNet ($N = 1.4 \cdot 10^7$, $d = 2 \cdot 10^5$). We also note that, if $N \gg d^2$, the NTK is low-rank and $\lambda_{\min}(K) = 0$. Furthermore, Corollary 4.1 and Theorem 4.2 cannot generally hold when $N \gg d^2$, as there are more data points to fit than parameters to help with the fitting. A milder requirement is possible for classification tasks, see the discussion at the end of Section 5. The second condition (5) is rather mild, as $\gamma$ can be taken to be arbitrarily small, and it avoids an exponential bottleneck in the last hidden layer.

**Notation.** The feature matrix at layer $l$ is $F_l = [f_l(x_1), \ldots, f_l(x_N)]^\top \in \mathbb{R}^{N \times n_l}$, and $\Sigma_l(x) = \mathrm{diag}([\phi'(g_{l,j}(x))]_{j=1}^{n_l})$ for $l \in [L-1]$, where $g_{l,j}(x)$ is the pre-activation neuron. Given two matrices $F, B \in \mathbb{R}^{m \times n}$, we denote by $F \circ B$ their Hadamard product, and by $F * B = [(F_{1:} \otimes B_{1:}), \ldots, (F_{m:} \otimes B_{m:})]^T \in \mathbb{R}^{m \times n^2}$ their row-wise Kronecker product (also known as Khatri-Rao product). Given a random vector $v$, let $\|v\|_{\psi_2}$ and $\|v\|_{\psi_1}$ denote its sub-Gaussian and sub-exponential norm, respectively (see also Appendix A for a detailed definition). Given a matrix $A$, let $\|A\|_{\mathrm{op}}$ be its operator norm, $\|A\|_F$ its Frobenius norm, $\lambda_{\min}(A)$ its smallest eigenvalue, and $\sigma_{\min}(A)$ its smallest singular value. We denote by $\|\varphi\|_{\mathrm{Lip}}$ the Lipschitz constant of the function $\varphi$. All the complexity notations $\Omega(\cdot)$, $\mathcal{O}(\cdot)$, $o(\cdot)$ and $\Theta(\cdot)$ are understood for sufficiently large $N, d, n_1, n_2, \ldots, n_{L-1}$. Tildes on such symbols are meant to neglect logarithmic factors.

## 3 NTK Bounds with Minimum Over-parameterization

Our main technical contribution on the smallest eigenvalue of the empirical NTK (3) is stated below.

**Theorem 3.1** (Smallest NTK eigenvalue under minimum over-parameterization). *Consider an $L$-layer neural network* (1), *where the activation function satisfies Assumption 2.3 and the layer widths satisfy Assumptions 2.4 and 2.5. Let $\{x_i\}_{i=1}^N \sim_{\text{i.i.d.}} P_X$, where $P_X$ satisfies the Assumptions 2.1-2.2,*

---

[3]By applying a Lipschitz map to a standard Gaussian distribution, the map output satisfies Assumption 2.2.

*and let $K$ be the empirical NTK Gram matrix (3). Assume that the weights of the network are initialized as $(W_l)_{i,j} \sim_{\text{i.i.d.}} \mathcal{N}(0, \beta_l^2/n_{l-1})$ for $l \in [L-1]$, $i \in [n_{l-1}]$, $j \in [n_l]$ and $(W_L)_i \sim_{\text{i.i.d.}} \mathcal{N}(0, \beta_L^2)$ for $i \in [n_{L-1}]$. Then, we have*

$$\lambda_{\min}(K) = \Omega(n_{L-2}n_{L-1}), \tag{6}$$

*with probability at least $1 - C\,Ne^{-c\log^2 n_{L-1}} - Ce^{-c\log^2 N}$ over $(x_i)_{i=1}^N$ and $(W_k)_{k=1}^L$, where $c$ and $C$ are numerical constants. Moreover, we have that*

$$\lambda_{\min}(K) = \mathcal{O}(dn_{L-1}), \tag{7}$$

*with probability at least $1 - Ce^{-cn_{L-1}}$, over $(x_i)_{i=1}^N$ and $(W_k)_{k=1}^L$.*

This result implies that the NTK is well conditioned for a class of networks in which the number of parameters $n_{L-2}n_{L-1}$ between the last two hidden layers is linear in the number of data samples $N$, up to logarithmic factors. This means that the total number of neurons of the network can be as little as $\tilde{\Omega}(\sqrt{N})$, which meets the *minimum possible amount of over-parameterization*. The lower bound (6) and the upper bound (7) match when all the widths (up to layer $L-2$) have the same scaling, i.e., $d = \Theta(n_{L-2})$. Throughout the paper, we do not explicitly track the dependence of our bounds on $L$, in the sense that the numerical constants $c, C$ may depend on $L$.

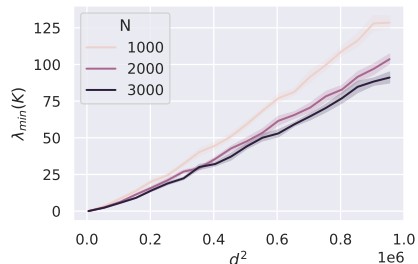

Figure 1: $\lambda_{\min}(K)$ as a function of $d^2$ in a 3-layer neural network, with sigmoid activation and $d = n_1 = n_2$.

In Figure 1, we consider a 3-layer neural network with $d = n_1 = n_2$, and we plot $\lambda_{\min}(K)$ as a function of $d^2$, for three different values of $N$. The inputs are sampled from a standard Gaussian distribution, the activation function is the sigmoid $\sigma(x) = (1 + e^{-x})^{-1}$, and we set $\beta_l = 1$ for all $l \in [L]$. We repeat the experiment 10 times, and report average and confidence interval at 1 standard deviation. The linear scaling of $\lambda_{\min}(K)$ in $d^2$ is in agreement with the result of Theorem 3.1. The code used to obtain the results of Figure 1 (and Figure 2 as well) is available at https://github.com/simone-bombari/smallest-eigenvalue-NTK/.

## 3.1 Roadmap of the proof of Theorem 3.1

After an application of the chain rule and some standard manipulations, we have

$$JJ^\top = \sum_{k=0}^{L-1} K_k, \qquad \text{with } K_k = F_k F_k^\top \circ B_{k+1}B_{k+1}^\top \in \mathbb{R}^{N \times N}, \tag{8}$$

where $B_k \in \mathbb{R}^{N \times n_k}$ is a matrix whose $i$-th row is given by

$$(B_k)_{i:} = \begin{cases} \Sigma_k(x_i)\left(\prod_{l=k+1}^{L-1} W_l \Sigma_l(x_i)\right) W_L, & k \in [L-2], \\ \Sigma_{L-1}(x_i) W_L, & k = L-1, \\ 1, & k = L. \end{cases} \tag{9}$$

From the decomposition (8), one readily obtains that $K \succeq K_{L-2}$, since $K_k$ is PSD for all $k \in \{0, \ldots, L-1\}$. Hence, it suffices to prove a lower bound on $\lambda_{\min}(K_{L-2})$. We denote by $J_{L-2}$ the Jacobian obtained by computing the gradient only over the parameters between layer $L-2$ and layer $L-1$. Then, $K_{L-2} = J_{L-2}J_{L-2}^\top$ and, by using (8) and (9), the $i$-th row of $J_{L-2}$ can be expressed as

$$(J_{L-2})_{i:} = f_{L-2}(x_i) \otimes (D_L\phi'(W_{L-1}^\top f_{L-2}(x_i))), \qquad i \in [N], \tag{10}$$

where $D_L$ is a diagonal matrix with the entries of $W_L$ on its diagonal. Note that, if we fix the weight matrices $(W_l)_{l=1}^L$, the rows $((J_{L-2})_{i:})_{i=1}^N$ are i.i.d., as the training data $(x_i)_{i=1}^N$ are i.i.d. too.

The proof consists of two main parts. First, we construct the centered Jacobian $\tilde{J}_{L-2}$, which is obtained from $J_{L-2}$ by iteratively removing expectations with respect to $X$ to its parts, and we show that its smallest singular value is *close* to the smallest singular value of $J_{L-2}$. The details of this part are contained in Appendix D. Second, we bound the smallest eigenvalue of the kernel obtained from

the centered Jacobian $\tilde{J}_{L-2}$ by providing an accurate estimate of the $\ell_2$ and sub-exponential norms of its rows. The details of this part are contained in Appendix E. In order to carry out this program, we exploit a number of concentration results on the $\ell_2$ norms of feature and backpropagation vectors, and on the $\ell_2$ norms of their centered counterparts. These results are contained in Appendix C, and they could be of independent interest. In particular, we provide tight high-probability estimates on *(i)* $\|f_l(x)\|_2$, *(ii)* $\mathbb{E}_x[\|f_l(x)\|_2^2]$, *(iii)* $\mathbb{E}_x[\|f_l(x)\|_2]$, *(iv)* $\mathbb{E}_x[\|f_l(x) - \mathbb{E}_x[f_l(x)]\|_2^2]$, $\mathbb{E}_x[\|f_l(x) - \mathbb{E}_x[f_l(x)]\|_2]$, and $\|f_l(x) - \mathbb{E}_x[f_l(x)]\|_2$, and *(v)* $\|D_L\phi'(g_{L-1}(x)) - \mathbb{E}_x[D_L\phi'(g_{L-1}(x))]\|_2$. Some preliminary calculations are also contained in Appendix B.

**Part 1: Centering.** We consider the centered Jacobian $\tilde{J}_{L-2}$, whose $i$-th row is defined as

$$(\tilde{J}_{L-2})_{i:} = \tilde{f}_{L-2}(x_i) \otimes (D_L\tilde{\phi}'(g_{L-1}(x_i))) - \mathbb{E}_{x_i}\left[\tilde{f}_{L-2}(x_i) \otimes D_L\tilde{\phi}'(g_{L-1}(x_i))\right], \quad i \in [N], \tag{11}$$

where $\tilde{f}_{L-2}(x_i) = \tilde{\phi}(g_{L-2}(x_i))$, with $\tilde{\phi}(g_{L-2}(x_i)) = \phi(g_{L-2}(x_i)) - \mathbb{E}_{x_i}[\phi(g_{L-2}(x_i))]$ and $\tilde{\phi}'(g_{L-1}(x_i)) = \phi'(g_{L-1}(x_i)) - \mathbb{E}_{x_i}[\phi'(g_{L-1}(x_i))]$ being the centered versions of the activation function and of its derivative, respectively. Our strategy is to relate $\lambda_{\min}(J_{L-2}J_{L-2}^\top)$ to $\lambda_{\min}(\tilde{J}_{L-2}\tilde{J}_{L-2}^\top)$ via the following two steps.

*Step (a): Centering $F_{L-2}$ and $B_{L-1}$.* We show that $\lambda_{\min}(J_{L-2}J_{L-2}^\top) \geq \lambda_{\min}(\tilde{J}_{FB}\tilde{J}_{FB}^\top) - o(n_{L-2}n_{L-1})$, where $\tilde{J}_{FB}\tilde{J}_{FB}^\top = \tilde{F}_{L-2}\tilde{F}_{L-2}^\top \circ \tilde{B}_{L-1}\tilde{B}_{L-1}^\top$ and $\tilde{F}_{L-2} = F_{L-2} - \mathbb{E}_X[F_{L-2}]$ and $\tilde{B}_{L-1} = B_{L-1} - \mathbb{E}_X[B_{L-1}]$ (Lemma D.1 in Appendix D.1). Our strategy differs from existing work (e.g., [61, 48]), where either only the backpropagation term is centered (cf. Proposition 7.1 of [61]) or only the features are centered (cf. Lemma 5.4 of [48]). We remark that, in order to handle deep networks with minimum overparameterization[4], the centering of $F_{L-2}$ and $B_{L-1}$ is crucially performed *at the same time*. In fact, by doing so, certain terms containing an eigenvalue which is negative and large in modulus suitably cancel. As a result, the difference between the original kernel $J_{L-2}J_{L-2}^\top$ and the centered one $\tilde{J}_{FB}\tilde{J}_{FB}^\top$ can be bounded by a matrix whose operator norm is $o(n_{L-2}n_{L-1})$.

*Step (b): Centering everything.* We show that $\lambda_{\min}(\tilde{J}_{FB}\tilde{J}_{FB}^\top) \geq \lambda_{\min}(\tilde{J}_{L-2}\tilde{J}_{L-2}^\top) - o(n_{L-2}n_{L-1})$, where the rows of $\tilde{J}_{L-2}$ are given by (11) (Lemma D.2 in Appendix D.2). The idea is to decompose the difference $\tilde{J}_{FB}\tilde{J}_{FB}^\top - \tilde{J}_{L-2}\tilde{J}_{L-2}^\top$ into a rank-1 matrix plus a PSD matrix. Then, in order to bound the operator norm of the rank-1 term, we leverage the general version of the Hanson-Wright inequality given by Theorem 2.3 in [1] (see the tail bound on quadratic forms that we provide in Lemma B.5). This strategy avoids whitening and dropping rows (as done in [61]) and, hence, appears to be better suited to the deep case.

The main result of this part is stated below, and it is proved by combining Lemmas D.1 and D.2.

**Theorem 3.2** (Jacobian centering). *Consider the setting of Theorem 3.1, and let the rows of $J_{L-2}$ and $\tilde{J}_{L-2}$ be given by (10) and (11), respectively. Then, we have*

$$\lambda_{\min}\left(J_{L-2}J_{L-2}^\top\right) \geq \lambda_{\min}\left(\tilde{J}_{L-2}\tilde{J}_{L-2}^\top\right) - o(n_{L-2}n_{L-1}), \tag{12}$$

*with probability at least $1 - C\exp(-c\log^2 n_{L-1}) - C\exp(-c\log^2 N)$ over $(x_i)_{i=1}^N$ and $(W_k)_{k=1}^L$, where $c, C$ are numerical constants.*

**Part 2: Bounding the centered Jacobian.** If we fix the weight matrices $(W_l)_{l=1}^L$, the rows of $\tilde{J}_{L-2}$ are i.i.d. vectors of the form $u \otimes v - \mathbb{E}[u \otimes v]$, where both $u$ and $v$ are centered. By exploiting this structure, in Appendix E.1 we bound the $\ell_2$ and sub-exponential norms of these rows. The $\ell_2$ norm is bounded in Lemma E.2, and this result relies on the tight estimates on the $\ell_2$ norms of centered features and centered backpropagation terms given by Lemma C.4 and C.5, respectively. The sub-exponential norm is bounded in Lemma E.3, and we use again the tail bound of Lemma B.5, which exploits the version of the Hanson-Wright inequality in [1]. At this point, the problem is reduced to bounding the smallest eigenvalue of a matrix such that its rows are i.i.d. and we have a good control on their $\ell_2$ and sub-exponential norms. This goal can be achieved via the following

---

[4]In contrast, [61] considers shallow networks, and [48] requires the existence of a layer with roughly $\Omega(N)$ neurons.

result, whose proof follows from Theorem 3.2 in [2] and it is presented for completeness in Appendix E.2.

**Proposition 3.3.** *Let $\tilde{J}$ be a matrix with rows $\tilde{J}_{i:} \in \mathbb{R}^{n_{L-2}n_{L-1}}$, for $i \in [N]$. Assume that $\{\tilde{J}_{i:}\}_{i \in [N]}$ are independent sub-exponential random vectors with $\psi = \max_i \|\tilde{J}_{i:}\|_{\psi_1}$. Let $\eta_{\min} = \min_i \|\tilde{J}_{i:}\|_2$ and $\eta_{\max} = \max_i \|\tilde{J}_{i:}\|_2$. Set $\xi = \psi K + K'$, and $\Delta := C\xi^2 N^{1/4}(n_{L-1}n_{L-2})^{3/4}$. Then, we have that*

$$\lambda_{\min}\left(\tilde{J}\tilde{J}^\top\right) \geq \eta_{\min}^2 - \Delta \tag{13}$$

*holds with probability at least*

$$1 - \exp\left(-cK\sqrt{N}\log\left(\frac{2n_{L-1}n_{L-2}}{N}\right)\right) - \mathbb{P}\left(\eta_{max} \geq K'\sqrt{n_{L-1}n_{L-2}}\right), \tag{14}$$

*where $c$ and $C$ are numerical constants.*

The main result of this part is stated below, and it is proved in Appendix E.3.

**Theorem 3.4** (Bounding the centered Jacobian). *Consider the setting of Theorem 3.1, and let the rows of $\tilde{J}_{L-2}$ be given by (11). Then, we have*

$$\lambda_{\min}\left(\tilde{J}_{L-2}\tilde{J}_{L-2}^\top\right) = \Omega(n_{L-2}n_{L-1}), \tag{15}$$

*with probability at least $1 - Ce^{-c\sqrt{N}} - CNe^{-c\log^2 n_{L-1}}$ over $(x_i)_{i=1}^N$ and $(W_k)_{k=1}^L$, where $c$ and $C$ are numerical constants.*

Recall that $K \succeq K_{L-2}$, hence $\lambda_{\min}(K) \geq \lambda_{\min}(K_{L-2}) = \lambda_{\min}\left(J_{L-2}J_{L-2}^\top\right)$. Thus, the lower bound (6) follows by combining the results from Theorem 3.2 and Theorem 3.4.

Finally, the upper bound (7) is rather straightforward. First, we write

$$\lambda_{\min}(K) = \lambda_{\min}\left(JJ^\top\right) \leq (JJ^\top)_{11} = \sum_{l=0}^{L-1} \|(F_l)_{1:}\|_2^2 \|(B_{l+1})_{1:}\|_2^2. \tag{16}$$

Then, by combining the bound on $\|(F_l)_{1:}\|_2^2$ of Lemma C.1 with a direct calculation for $\|(B_{l+1})_{1:}\|_2^2$, the desired result (7) follows. The details are contained in Appendix F.

## 4 Two Applications: Memorization and Optimization

**Memorization capacity.** The fact that the NTK Gram matrix (3) is well conditioned readily implies a result on memorization capacity. This was already observed in [42] for two-layer networks and in [48] for deep networks with a layer containing $\Omega(N)$ neurons. Here, by using Theorem 3.1, we show that $\Omega(N)$ parameters between the last pair of hidden layers are enough for the network to fit $N$ real-valued labels up to an arbitrarily small error. The result is stated below and, for completeness, we give the proof in Appendix G.

**Corollary 4.1** (Memorization). *Consider an $L$-layer neural network (1), where the activation function satisfies Assumption 2.3 and the layer widths satisfy Assumptions 2.4 and 2.5. Let $\{x_i\}_{i=1}^N \sim_{\text{i.i.d.}} P_X$, where $P_X$ satisfies the Assumptions 2.1-2.2. Then, it holds that for every $Y \in \mathbb{R}^N$, and for every $\varepsilon > 0$, there exists a set of parameters $\theta$ such that*

$$\|F_L(\theta) - Y\|_2 \leq \varepsilon, \tag{17}$$

*with probability at least $1 - CNe^{-c\log^2 n_{L-1}} - Ce^{-c\log^2 N}$ over $(x_i)_{i=1}^N$, where $c$ and $C$ are numerical constants.*

**Gradient descent training.** Theorem 3.1 has implications on the convergence of gradient descent algorithms. In particular, by choosing carefully the initialization, we show that gradient descent trained on the $N$ samples $\{x_i\}_{i=1}^N$ with labels $Y = (Y_1, \ldots, Y_N)$ converges to zero loss. This is – at the best of our knowledge – the *first result of this kind for deep networks with minimum over-parameterization.*

To define our initialization, let us assume that the $(L-1)$-th layer has an even number of neurons. With a slight abuse of notation, we indicate its width as $2n_{L-1}$. Thus, $W_{L-1}$ can be written in the

form $\left[W_{L-1}^{(1)}, W_{L-1}^{(2)}\right]$, where $W_{L-1}^{(1)}, W_{L-1}^{(2)} \in \mathbb{R}^{n_{L-2} \times n_{L-1}}$. Similarly, $W_L \in \mathbb{R}^{2n_{L-1}}$ is the concatenation of $W_L^{(1)}, W_L^{(2)} \in \mathbb{R}^{n_{L-1}}$. Then, we define the initialization $\theta_0 = [\text{vec}(W_{1,0}), \dots, \text{vec}(W_{L,0})]$ as follows:

$$(W_{l,0})_{i,j} \sim_{\text{i.i.d.}} \mathcal{N}(0, \beta_l^2/n_{l-1}), \ l \in [L-2], \ i \in [n_{l-1}], \ j \in [n_l],$$

$$(W_{L-1,0}^{(1)})_{i,j} \sim_{\text{i.i.d.}} \mathcal{N}(0, \beta_{L-1}^2/n_{L-2}), \ i \in [n_{L-2}], \ j \in [n_{L-1}], \text{ and } W_{L-1,0}^{(2)} = W_{L-1,0}^{(1)}, \quad (18)$$

$$(W_{L,0}^{(1)})_i \sim_{\text{i.i.d.}} \mathcal{N}(0, \beta_L^2\gamma), \ i \in [n_{L-1}], \text{ and } W_{L,0}^{(2)} = -W_{L,0}^{(1)}.$$

As usual, $(\beta_l)_{l=1}^L$ are numerical constants independent of the layer widths. In contrast, the quantity $\gamma$ will be set to a suitably large value depending on the number of training samples and on the layer widths. In words, the initialization is standard for $W_{1,0}, \dots, W_{L-2,0}$ and $W_{L-1,0}^{(1)}$; the variance of $W_{L,0}^{(1)}$ is boosted up by a factor $\gamma$; and we duplicate the neurons of layer $L-1$ so that $W_{L-1,0}^{(2)} = W_{L-1,0}^{(1)}$ and $W_{L,0}^{(2)} = -W_{L,0}^{(1)}$. At this point, we are ready to state our optimization result.

**Theorem 4.2** (Optimization via gradient descent). *Consider an $L$-layer neural network* (1)*, where the activation function satisfies Assumption 2.3 and the layer widths satisfy Assumptions 2.4 and 2.5. Consider training data $\{x_i\}_{i=1}^N \sim_{\text{i.i.d.}} P_X$, where $P_X$ satisfies the Assumptions 2.1-2.2, with labels $Y \in \mathbb{R}^N$ such that $\|Y\|_2 = \Theta(\sqrt{N})$. We consider solving the least-squares optimization problem*

$$\min_\theta \mathcal{L}(\theta) := \frac{1}{2} \min_\theta \|F_L(\theta) - Y\|_2^2, \quad (19)$$

*by running gradient descent updates of the form $\theta_{t+1} = \theta_t - \eta \nabla \mathcal{L}(\theta_t)$, where the initialization $\theta_0$ is defined in* (18) *with $\gamma = d^3 N^2$ and $\eta \le C(\gamma N d n_{L-1})^{-1}$. Then, for all $t \ge 1$,*

$$\mathcal{L}(\theta_t) \le (1 - c\eta\gamma n_{L-2} n_{L-1})^t \mathcal{L}(\theta_0), \quad (20)$$

*with probability at least $1 - C N e^{-c \log^2 n_{L-1}} - C e^{-c \log^2 N}$ over $(x_i)_{i=1}^N$ and the initialization $\theta_0$, where $c$ and $C$ are numerical constants.*

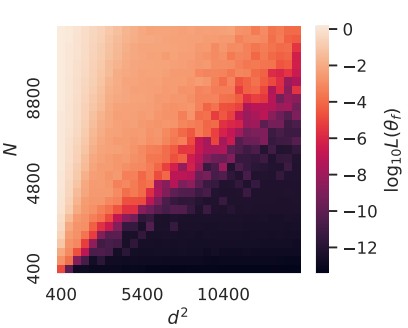

Figure 2: Value of the loss $\mathcal{L}(\theta_f)$ once gradient descent has converged for a 4-layer ReLU network with $d = n_1 = n_2 = n_3$.

In words, (20) guarantees that gradient descent *linearly* converges to a point with zero loss. We also note that (our upper bound on) the convergence rate can be expressed as a simple function of the learning rate $\eta$, the number of training samples $N$, and the layer widths $\{n_i\}_{i=0}^{L-1}$.

In Figure 2, we give an illustrative example that 4-layer networks achieve 0 loss when the number of parameters is at least linear in the number of training samples, i.e., under minimum over-parameterization. To ease the experimental setup, we use a ReLU activation, with Adam optimizer. We initialize the network as in the setting of Theorem 3.1, picking $\beta_l = 1$ for all $l \in [L]$. The inputs, as well as the targets, are sampled from a standard Gaussian distribution. The plot is averaged over 10 independent trials. As predicted by Theorem 4.2, the loss experiences a phase transition and it goes from 1 to 0 when the layer widths are of order $\sqrt{N}$.

We now provide a sketch of the argument of Theorem 4.2, deferring the detailed proof to Appendix H. As a consequence of Theorem 3.1, the NTK is well conditioned at the initialization $\theta_0$, which implies that gradient descent makes progress at the beginning. The idea is that the NTK remains sufficiently well conditioned also during training and, hence, the loss vanishes with the linear rate in (20), since the trained weights remain confined in a ball of sufficiently small radius centered at $\theta_0$. More specifically, we exploit the optimization framework developed in [49]. There, a convergence result is proved if the following condition holds: there exists $\alpha, \beta$ such that

$$\alpha \le \sigma_{\min}(J(\theta)) \le \|J(\theta)\|_{\text{op}} \le \beta, \quad \text{and} \quad \|J(\theta_1) - J(\theta_2)\|_{\text{op}} \le \frac{\alpha^2}{2\beta}, \quad (21)$$

for all $\theta, \theta_1, \theta_2 \in \mathcal{D} = \mathcal{B}(\theta_0, R)$, where $\mathcal{D}$ denotes a ball centered at $\theta_0$ of radius $R := 4\|F_L(\theta_0) - Y\|_2/\alpha$ (cf. Proposition H.1). In order to control the radius $R$, we take advantage of the form (18) of the initialization $\theta_0$. In particular, the spectrum of the Jacobian $J$ (and, hence, $\alpha$) scales linearly in $\sqrt{\gamma}$, and the network output at initialization $F_L(\theta_0)$ is equal to 0 (cf. Lemma H.2). Thus, as $\|Y\|_2 = \Theta(\sqrt{N})$, we have that $R = \Theta(\sqrt{N}/\alpha)$, which can be controlled by choosing suitably $\gamma$. At this point, the argument consists in providing a number of estimates on feature vectors, backpropagation terms, and finally the Jacobian $J(\theta)$ for all $\theta \in \mathcal{D}$ (cf. Lemmas H.3-H.9). This allows us to find $\alpha, \beta$ such that (21) holds and conclude.

## 5 Related Work and Discussion

**Random matrices in deep learning.** The limiting spectra of several random matrices related to neural networks have been the subject of a recent line of work. In particular, the Conjugate Kernel (CK) – namely, the Gram matrix of the features from the last hidden layer – has been studied for models with two layers [54, 37, 39, 51], with a bias term [3, 56], and with multiple layers [13]. The Hessian matrix has been considered in [52], and the closely related Fisher information matrix in [55]. Using tools from free probability, the input-output Jacobian (as opposed to the parameter-output Jacobian considered in this work) of deep networks is studied in [53] and the NTK of a two-layer model in [4]. In [23], the spectrum of NTK and CK for deep networks is characterized via an iterated Marchenko-Pastur map. The generalization error has also been studied via the spectrum of suitable random matrices, see [29, 41, 38, 42] and Section 6 of the review [11]. Most of the existing results focus on the linear-width asymptotic regime, where the widths of the various layers are linearly proportional. An exception is [68], which focuses on the ultra-wide two-layer case ($n_1 \gg N$).

**Smallest eigenvalue of empirical kernels.** In the line of work mentioned above, the limiting spectrum of the random matrix is characterized in terms of weak convergence, which does not lead to a direct implication on the behavior of its smallest eigenvalue. In fact, lower bounding the smallest eigenvalue often requires understanding the speed of convergence to the limit spectrum. Quantitative bounds for random Fourier features are obtained in [9]. In the two-layer setting, the smallest NTK eigenvalue is lower bounded in [61, 42, 68], and concentration bounds can also be obtained from [4, 63, 31]. In particular, [42] establishes a tight bound for two-layer networks with roughly as many parameters as training samples (hence, under minimum over-parameterization), and for a wide class of activations. However, this result still requires that $n_1 = \Omega(d)$. For deep networks, a convergence rate on the NTK can be obtained from [7] and potentially from [15], however these results require all the layers to be wide. In [48], it is proved that a single layer of linear width suffices for the NTK to be well conditioned. Here, Theorem 3.1 provides the first result for deep networks under minimum over-parameterization, therefore allowing for sub-linear layer widths.

**Memorization capacity and gradient descent training.** For binary classification, the memorization capacity of neural networks was first studied by Cover [18], who solved the single-neuron case. Later, Baum [12] proved that, for two-layer networks, the memorization capacity is lower bounded by the number of parameters, and upper bounds of the same order were given in [35, 58]. Recently, tight results for two-layer networks in the regression setting have been provided [14, 42]. In particular, [14] generalizes the construction of [12], while the memorization result of [42] comes from a bound on the smallest NTK eigenvalue. This same route is also followed to analyze deep networks in [48] and in this work. Results for multiple layers have been provided in [10, 71, 66, 24]. As concerns efficient algorithms achieving memorization, [19, 20] study classification with two-layer networks. A popular line of research has exploited the NTK view to give optimization guarantees on gradient descent. Existing work has focused on the two-layer case [61, 22, 50, 63, 69, 62], and optimal results in terms of over-parameterization have been obtained [42]. For deep networks, [21] assumes a lower bound on the smallest NTK eigenvalue, [5, 73, 74] require all the layer widths to be (rather large) polynomials in the number of samples, and [47, 43] reduce the over-parameterization to a single layer of linear width. In particular, in these last two papers, the analysis of the NTK is reduced to that of a certain feature matrix (the first one in [47] and the last one in [43]). Hence, these approaches do not seem suitable to tackle networks with sub-linear layer widths[5]. Finally, we remark that milder over-paramerization requirements are sufficient for classification problems. More specifically, it has been proved that polylogarithmic width suffices for two-layer [34] and deep [16]

---

[5]If the width is sub-linear, then the feature matrix is low-rank and, hence, its smallest eigenvalue is 0.

ReLU networks to converge and generalize. In fact, achieving memorization for regression is harder than for classification, where it suffices to satisfy inequality constraints [42].

# 6 Concluding Remarks

In this paper, we show that the NTK is well conditioned for deep networks with sub-linear layer widths. As consequences of our NTK bounds, we also prove results on memorization and optimization for such a class of networks. Our approach requires smooth activations and layer widths which do not significantly grow in size with the depth. We believe such assumptions to be purely technical, and we leave as an open question their removal. Our novel NTK analysis could potentially be applied beyond standard gradient descent, e.g., for optimization with momentum [67], or federated learning [32].

## Acknowledgements

The authors were partially supported by the 2019 Lopez-Loreta prize, and they would like to thank Quynh Nguyen, Mahdi Soltanolkotabi and Adel Javanmard for helpful discussions.

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
