# OpenReview forum: "Memorization and Optimization in Deep Neural Networks with Minimum Over-parameterization"
_NeurIPS.cc/2022/Conference — NeurIPS 2022 Accept_

### Official Review · Reviewer_mvcT · 2022-07-11

**Rating:** 5
**Confidence:** 4
**Soundness:** 3 good
**Presentation:** 3 good
**Contribution:** 3 good

**Summary:**

This paper shows that NTK can be well-conditioned in a sub-linear setup. In particular, they proved that roughly  $\Omega (N)$ parameters are needed to get a good NTK matrix, and also $O(N)$ parameters are enough to memorize the training data. Based on those NTK bounds, they further provide results on the memorization capacity and the optimization via gradient descent and square loss.

**Questions:**

1. Requirement 4 in Assumption 2.5 implies that the total number of neurons should be larger than $\Omega(N^{(1+\alpha)/2})$. This violates the statement after Theorem 3.1 in line 175.

2. The authors argue that when all the widths have the same scaling, the upper bound matches the lower bound. In this case, the training data size $N$ should be smaller than $d^{2/(1+\alpha)}$ according to Assumption 2.5, where d is the dimension of the input. This is a fairly strong condition for the training data set, which may limit the significance of the results.

3. Under certain margin assumptions on the training data, [1] showed that a polylogarithmic width condition suffices for two-layer ReLU networks to converge and generalize. Later [2] studied how much over-parameterization is sufficient to learn deep ReLU networks and showed that under certain assumptions on the training data, polylogarithmic width is sufficient. Since these papers have milder overparameterization conditions of the NNs, the authors may want to make a comparison with those results.

[1] Ji, Ziwei, and Matus Telgarsky. "Polylogarithmic width suffices for gradient descent to achieve arbitrarily small test error with shallow ReLU networks." International Conference on Learning Representations. 2019.

[2] Chen, Zixiang, et al. "How Much Over-parameterization Is Sufficient to Learn Deep ReLU Networks?." International Conference on Learning Representations. 2020.

**Limitations:**

The authors have addressed their work's limitations and potential negative social impact.

**Strengths And Weaknesses:**

It is interesting to study the neural network in a sub-linear setup. For square loss, the authors provided valuable sight for the training data memorization as well as the optimization. This paper is well-written and clear for the technique part. However, I have several concerns about the significance and the limitation of the results. Besides,  I think this paper can improve the clarity of their results by making a comparison with the setting for the classification.

---

> ### Author Response · Authors · 2022-08-01
> **Response to Reviewer mvcT**
>
> We thank the reviewer for the detailed comments. We reply below to the three questions posed by the reviewer, and we are happy to answer additional follow-up questions. We hope that our response solves the concerns about the significance of our work. If this is the case, we also hope that the reviewer will consider raising the rating.
>
> ----
>
> **Question 1.**
>
> We believe there is a misunderstanding here, and we would like to clarify this point. The first requirement of Assumption 2.5 is that there exists $\alpha>0$ such that (4) holds. First, notice that choosing a *larger* $\alpha$ leads to a *more restrictive* condition (4). As (4) is required to hold only *for some* $\alpha>0$, we are allowed to pick this $\alpha$ to be very small, e.g. $\alpha=0.001$.
>
> In fact, the *tight* over-parameterization condition would be $N=o(n_{L-2}n_{L-1})$. The point of (4) is to allow a small slack in the exponent via the additional parameter $\alpha$, in order to take care of extra logarithmic factors appearing in the argument. More specifically, by tracking such logarithmic factors, the requirement (4) in Assumption 2.5 can be replaced by $N\log^{12} N=o(n_{L-2}n_{L-1})$ (which now does not contain $\alpha$ anymore). We have refrained from doing that in the submitted version, since we deemed (4) as written more clean and clear. However, we acknowledge that the current writing has led to a question by both this reviewer and reviewer *UvzJ* (see also the first point in the corresponding response). Hence, if the reviewers find it useful, we are happy to replace the requirement $N^{1+\alpha}=o(n_{L-2}n_{L-1})$ with the slightly less restrictive requirement $N\log^{12} N=o(n_{L-2}n_{L-1})$ and perform the necessary (very minor) edits to the proofs in the appendix.
>
> As a final note, in the revision, we have changed the statement
>
> “This means that the total number of neurons of the network can be as little as $\Omega(\sqrt{N})$"
>
> into
>
> “This means that the total number of neurons of the network can be *roughly* as little as $\Omega(\sqrt{N})$”,
>
> where the *roughly* captures the extra logarithmic factors (or small slack in the exponent), see l. 176 of the revision.
>
> ----
>
> **Question 2.**
>
> Following our response to the previous point, we can take $\alpha$ to be arbitrarily small and, hence, the condition mentioned by the reviewer is that $N=o(d^2)$ (up to extra logarithmic factors). We would like to make two comments about this condition.
>
> The first comment is theoretical in nature. When all the widths have the same scaling, then having that $N$ scales at most as $d^2$ is **necessary** for the results of our paper to hold. In fact, if $N$ scales faster than $d^2$, the NTK is a low-rank matrix and, therefore, its smallest eigenvalue is $0$. The two consequences of our main result (Corollary 4.1 about memorization and Theorem 4.2 about optimization) are also not true in general when $N$ scales faster than $d^2$. In fact, in this setting, there are more data points to fit than parameters to help with the fitting. This argument can be made formal by bounding the VC dimension, see e.g. [10].
>
> The second comment is more linked to the practice of neural networks trained on standard datasets. Most practical datasets are high-dimensional and this is also true for some of the most ‘standard’ ones: for MNIST, $N=60000$ and $d=784$, hence $N\ll d^2 \approx 6\cdot 10^5$; for CIFAR-10, $N=5\cdot 10^4$ and $d=3\cdot 32^2$, hence $N\ll d^2 \approx 10^7$; for ImageNet, $N=1.4\cdot 10^7$ and $d=2\cdot 10^5$, hence $N\ll d^2 \approx 4\cdot 10^{10}$. Thus, it is reasonable to say that the condition $N=o(d^2)$ is satisfied in (some) realistic settings. We have added this observation in l. 180-185 of the revision.
>
> For another discussion about the significance of our work, we refer the reviewer to our answer to the first question of reviewer *5UpM*.
>
> ----
>
> **Question 3.**
>
> This is an excellent suggestion. Let us point out that achieving memorization (and, similarly, performing optimization) for regression is harder than for classification. In fact, achieving memorization for regression means to satisfy *exact equalities*, while for classification it suffices to satisfy *inequality constraints*. This is the fundamental reason why the number of parameters has to be at least linear in the number of data points for regression, while polylogarithmic widths are sufficient for classification. We have clarified this point and we have added the two references suggested by the reviewer at the end of Section 5 of the revision.

---

> > ### Comment · Reviewer_mvcT · 2022-08-07
> > **Follow up comment**
> >
> > Thanks to the reviewers for the detailed response. I am satisfied with the author's comments on the classification problem and agree with the reviewer that achieving memorization (and, similarly, performing optimization) for regression is harder than for classification. Therefore, I will increase my score from 4 to 5.
> >
> > 1. I am convinced that most of the standard datasets satisfy the condition $n < d^{2/(1+\alpha)}$. However, since the $n > d^{2}$ case is a more challenging problem for memorization, I would like to suggest the author make it clear in the assumption section rather than after the theorem.
> >
> > 2. "Roughly" make the statements in the paper vague and hard to follow. I think the authors should replace them with rigorous conditions. It would be great if the authors could avoid introducing the term $\alpha$. In that case, the authors can claim that the total number of neurons in the network can be as little as $\tilde{\Omega}(\sqrt{N})$.

---

> > > ### Author Response · Authors · 2022-08-08
> > > **Follow up response and revision**
> > >
> > > We thank reviewer *mvcT* for the constructive comments and for raising the score. We have uploaded a slightly edited revision which incorporates the follow-up comment.
> > >
> > > The main change is to replace the requirement (4) in Assumption 2.5 by $N\log^{8} N=o(n_{L-2}n_{L-1})$. This last expression does not contain the parameter $\alpha$ anymore and, after carefully tracking the logarithms, we have also been able to improve upon the requirement mentioned in our previous response ($N\log^{12} N=o(n_{L-2}n_{L-1})$).
> > >
> > > The changes to the arguments are minor and we summarize them below:
> > >
> > > * We have changed (22) and (25) in Lemma B.1, and updated the proof of this lemma since it uses the new over-parameterization requirement $N\log^{8} N=o(n_{L-2}n_{L-1})$.
> > >
> > > * Lemma B.1 is used twice in the proof of Lemma D.1 (l. 1066 and l. 1075) and once in the proof of Lemma D.2 (l. 1107). These parts do not require any change since we can simply apply the new version of Lemma B.1.
> > >
> > > * Lemma B.1 is used in the proof of Theorem 3.4 contained in Appendix E.3, and the lines 1173-1175 have been slightly edited to reflect the change in Lemma B.1.
> > >
> > > The rest of the argument is not affected by our change in the requirement (4).
> > >
> > > As regards the body of the paper, we have followed the reviewer’s suggestion and we have substituted the vague statements (containing “roughly”) with rigorous ones. In particular, we have edited l. 11-12 of the abstract, l. 52-54 in the introduction, l. 165-166 in the notation, and l. 179-180 after Theorem 3.1. Following the reviewer’s suggestion, we have also moved the discussion on the assumption $N=\tilde o(d^2)$ directly after Assumption 2.5, see l. 149-154, instead of having it after the statement of Theorem 3.1.
> > >
> > > We hope that the reviewer is fully satisfied with this revision, and that all concerns have been resolved. If this is not the case, we are happy to answer follow-up questions.

---

### Official Review · Reviewer_mnfH · 2022-07-11

**Rating:** 6
**Confidence:** 2
**Soundness:** 4 excellent
**Presentation:** 3 good
**Contribution:** 3 good

**Summary:**

For an $L$-layer (loosely) pyramidal network, this paper gives a probabilistic lower bound on the smallest eigenvalue of the empirical neural tangent kernel Gram matrix, which is proportional to the product of the width of the last two hidden layers. When combined with known results on NTK, this lower bound implies that $\sim \sqrt{N}$ neurons are enough to learn a solution (via GD) that memorizes $N$ copies of data.

**Questions:**

My questions are basically the same as the weaknesses described above:

- Could you describe what the hidden terms of Theorem 3.1. are?

- Please explain when and why $CN\exp(-c\log^2 n_{L-1})$ would be small enough for event in Theorem 3.1 would happen with a sufficiently large probability.

**Limitations:**

Limitations are well-addressed.

**Strengths And Weaknesses:**

I do not have anything major against the acceptance of this paper. The paper is well written and the theoretical contributions are clear (up to my knowledge). I especially appreciate the proof-technical innovation in the paper---the three-stage centering argument may be quite useful for the future works in this direction.

Nevertheless, I do have some minor concerns:

- Assumption 2.4 restricts the setup to neural networks with "loose pyramidal topology," which seems to be much weaker than the strict pyramidal topology in the sense that the layer width need only be smaller than a constant times the preceding layer's width. While this looks very easy to satisfy, this also suggests that actually many "constants" are being hidden inside the big-O notation of Theorem 3.1. Could you briefly describe the what the hidden terms of Theorem 3.1 are, and how they behave with respect to the assumption?

- The probability of event at Theorem 3.1. is $1 - C \exp(-c\sqrt{N}) - CN\exp(-c\log^2 n_{L-1})$. I am particularly worried about the last term, which is proportional to the sample size $N$. Could you explain in more detail, why, and under what condition such $N$-proportionality does not really make the probabilistic bound meaningless?

- I do not find Figure 1 to be very meaningful---any curve can look linear when we select a nice interval.

---

> ### Author Response · Authors · 2022-08-01
> **Response to Reviewer mnfH**
>
> We would like to thank the reviewer for the comments and for acknowledging the technical innovations introduced by our contribution. We reply below to the three (minor) concerns raised in the review, and we are happy to answer additional follow-up questions. In case the reviewer is satisfied with our response, we hope that raising the rating will be considered.
>
> ----
>
> **Concern 1**
>
> This is an excellent question. Let us discuss how the “hidden constants” affect the lower bound (6), which constitutes the key result of this paper, and the crucial over-parameterization requirement (4) needed for such a result to hold.
>
> As for the lower bound in (6), the $\Omega$ in the LHS does not hide any terms depending on the network topology. To see this, note that the smallest eigenvalue of the NTK is lower bounded by the smallest eigenvalue of the centered NTK via Theorem 3.2, and the difference between these two quantities is $o(n_{L-2}n_{L-1})$. Furthermore, by examining (213) in Appendix E.3, we have that the lower bound on the smallest eigenvalue of the centered NTK comes from the $\ell_2$ norm of the rows of the centered Jacobian minus some terms which are again $o(n_{L-2}n_{L-1})$ (the quantity $\Delta$ is bounded in Eq. (212)). As the $\ell_2$ norm of the rows of the centered Jacobian does not depend on the network topology with the practical initialization considered in Theorem 3.1, also the lower bound (6) is not affected by the topology. We remark that this $\ell_2$ norm and, therefore, the lower bound (6) both depend on the Lipschitz constant of the activation function.
>
> As for the over-parameterization requirement (Eq. (4) in Assumption 2.5), we note that this would include a term of the form $(\prod_l \max(1, n_l/n_{l-1}))^{16}$. This can be noticed by examining Eq. (212) in Appendix E.3. In fact, in order to ensure that $\Delta$ is $o(n_{L-2}n_{L-1})$, it suffices that $N$ scales with the sub-exponential norm of the rows of the centered Jacobian raised to the 8-th power. The sub-exponential norm of the rows of the centered Jacobian scales as the square of the Lipschitz constant of the feature map, and this Lipschitz constant scales as $\prod_l \max(1, n_l/n_{l-1})$. We remark that this dependence on the network topology is not tight, and we believe it could be improved. We have not pursued this direction so far, in order to make our assumptions easier to grasp and the proofs easier to follow.
>
> ----
>
> **Concern 2**
>
> After the edits in the argument in Appendix D, the probability appearing in Theorem 3.1 has been changed to $1 - C N e^{-c \log^2 n_{L-1}} - C e^{-c \log^2 N}$. However, this change should not impact the concern of the reviewer, which is about the term $C N e^{-c \log^2 n_{L-1}}$. In fact, it is clear that the term $C e^{-c \log^2 N}$ vanishes for large $N$.
>
> Let us recall that, by Eq. (5) in Assumption 2.5, $n_{L-1}$ cannot be exponentially smaller than $N$. In particular, we require the existence of $\gamma>0$ such that $N^\gamma = \mathcal O(n_{L-1})$. This assumption avoids exponential bottlenecks in the neural network and, as a consequence, it also ensures that the term considered by the reviewer vanishes. In fact, this term can be bounded as follows:
>
> $CN e^{-c \log^2 n_{L-1}} \leq CN e^{−\gamma^2 c \log^2 ⁡N} =  C e^{\log N − \gamma^2 c \log^2 ⁡N},$
> which goes to $0$ as $N$ grows.
>
> In general, the term $CN e^{-c \log^2 n_{L-1}}$ vanishes as long as the width $n_{L-1}$ is not exponentially small when compared to the dataset size $N$. This last assumption is mild and, for concreteness, we also consider a simple numerical example. Consider the ImageNet dataset ($N=1.4\cdot 10^7$), assume all layer widths to be equal to the input dimension ($n_{L-1} = d = 2\cdot 10^5$), and set all constants to 1. Then, we obtain $N e^{ - \log^2 n_{L-1}}=1.4 \cdot 10^7 e^{- (\log(2 \cdot 10^ 5))^2} \approx 10^{-58}$, which is very small, as predicted.
>
> ----
>
>
> **Concern 3**
>
> We agree with the reviewer that any curve looks linear in a sufficiently small neighborhood of any given point. However, we would like to remark that the smallest eigenvalue of the NTK exhibits a linear behavior in $d^2$ for a rather large interval of values of $d^2$. In fact, we have repeated the same experiment for $N=3000$ after increasing by a factor $4$ the values in the x-axis of the previous plot, and we have obtained very similar results. The new version of the plot with larger values on the x-axis is contained in the file `scaling_wider.pdf` that can be found in the supplementary material of the revision.

---

### Official Review · Reviewer_UvzJ · 2022-07-12

**Rating:** 6
**Confidence:** 1
**Soundness:** 3 good
**Presentation:** 3 good
**Contribution:** 3 good

**Summary:**

This submission studies the problem of minimum over-parameterization in deep neural networks. More specifically, the authors investigated the settings with sublinear layer width and provides the lower bound in terms of the smallest NTK eigenvalue.

**Questions:**

See above

**Limitations:**

Yes

**Strengths And Weaknesses:**

I must first say that this paper is much beyond my expertise, and the evaluation of this submission should depend on other reviewers with necessary background.

Strength:
As the authors claimed, this paper gives an affirmative answer regarding whether NTK is well-conditioned in the sublinear setting. The theoretical results from Theorem 3.1 match the claim. The authors also provide details proof to cover the main steps.

Weakness:
Some clarification regarding the assumptions would be helpful to understand the major conclusion. For example, if \alpha >= 1 in Eq. (4), does it mean that the number of neurons would exceed the sublinear case?

Another case is regarding the proof for the smallest NTK eigenvalue: how does it depend on the optimizer? Since NTK changes with updated parameters (which may explode if optimization failed), does the proof depend on any hidden assumptions for the optimizer?

---

> ### Author Response · Authors · 2022-08-01
> **Response to Reviewer UvzJ**
>
> We thank the reviewer for the comments. We clarify below the two points raised as ‘Weaknesses’, and we are happy to answer additional follow-up questions.
>
> ----
>
> **Question about $\alpha \ge 1$**
>
> We believe there is a misunderstanding here, and we would like to clarify this point. The first requirement of Assumption 2.5 is that *there exists* $\alpha>0$ such that (4) holds. First, notice that choosing a *larger* $\alpha$ leads to a *more restrictive* condition (4). As (4) is required to hold only *for some* $\alpha>0$, we are allowed to pick this $\alpha$ to be very small, e.g. $\alpha=0.001$. Thus, the case $\alpha\ge 1$ does not lead to the desired minimum over-parameterization requirement.
>
> In fact, the *tight* over-parameterization condition would be $N=o(n_{L-2}n_{L-1})$. The point of (4) is to allow a small slack in the exponent via the additional parameter $\alpha$, in order to take care of extra logarithmic factors appearing in the argument. More specifically, by tracking such logarithmic factors, the requirement (4) in Assumption 2.5 can be replaced by $N\log^{12} N=o(n_{L-2}n_{L-1})$ (which now does not contain $\alpha$ anymore). We have refrained from doing that in the submitted version, since we deemed (4) as written more clean and clear. However, we acknowledge that the current writing has led to a question by both this reviewer and reviewer *mvcT* (see also the first point in the corresponding response). Hence, if the reviewers find it useful, we are happy to replace the requirement $N^{1+\alpha}=o(n_{L-2}n_{L-1})$ with the slightly less restrictive requirement $N\log^{12} N=o(n_{L-2}n_{L-1})$ and perform the necessary (very minor) edits to the proofs in the appendix.
>
> **[EDIT]** After the follow-up comment of reviewer *mvcT*, we have implemented this change and further improved the over-parameterisation requirement to $N\log^{8} N=o(n_{L-2}n_{L-1})$. The changes to the arguments in the appendix and to the body of the paper are minor, and they are detailed in the follow-up response to reviewer *mvcT*.
>
> ----
>
> **Question regarding the optimization**
>
> We think the reviewer is referring to the optimization result (Theorem 4.2). If this is not the case and we misinterpreted the comment, please let us know and we are happy to clarify this point further.
> The reviewer is correct when mentioning that the NTK changes as the parameters get updated and, hence, one needs to make sure that such parameters do not grow unbounded. In fact, this is precisely our proof strategy for Theorem 4.2. The core of the argument consists in controlling how the NTK changes along the optimization trajectory. More specifically, we bound the radius of a ball that contains the whole trajectory of gradient descent (from initialization until a point with 0 loss is reached), and we also bound the NTK spectrum inside this ball. The detailed proof is provided in Appendix H, and there are no hidden additional assumptions.

---

### Official Review · Reviewer_5UpM · 2022-07-13

**Rating:** 6
**Confidence:** 2
**Soundness:** 3 good
**Presentation:** 2 fair
**Contribution:** 2 fair

**Summary:**

This manuscript analyses the neural tangent kernel in the regime of minimum over-parameterization. In detail, the authors prove the lower bound about the NTK eigenvalue for deep nets with minimum over-parameterization. Then, they apply the theorems to the aspects of networks' memorization and training.

**Questions:**

1. The modern deep learning foucses on "super" over-parameterization, why do the paper research the minimum over-parameterization? For completing the NTK theorem? How the research on the minimum over-parameterization help understand the success of deep nets?

2. What is the mean of "to be conditioned"?

3. About the experiments of Figure 2, are the used labels random? If yes, Can we get the similar results with normal label (or normal dataset with low complexity, compared to the complex random datasets)?


**Limitations:**

Please the Questions.

**Strengths And Weaknesses:**

Pros:

This manuscript deeply investigate the question of minimum over-parameterization;

The proofs are solid;

Two application are proposed based on theorems.

Cons:

This paper is very theoretical and focuses on the minimum over-parameterization in deep learning. I am not an expert about the NTK research, so I mainly give some comments and suggestion on writting. For me, it is neccesary to clearly and comprehensively illustrate the backgroud and significance on investigating the problem of minium over-parameterizatio, please see Questions below.

Some minor writing suggestions:

1. Line 60: the full name of PSD should be proposed;

---

> ### Author Response · Authors · 2022-08-01
> **Response to Reviewer 5UpM**
>
> We thank the reviewer for the feedback. As for the minor writing suggestion on the PSD acronym, we have corrected the paper accordingly. We reply below to the three questions posed by the reviewer, and we are happy to answer additional follow-up questions. In case the reviewer is satisfied with our response (especially the discussion about the significance of proving guarantees under minimum over-paramerization), we hope that raising the rating will be considered.
>
> ----
>
> **Question 1**
>
> While it is certainly true that modern deep learning models are vastly over-parameterized, in the era of “big data” the sizes of datasets have also exploded. This means that it is not really realistic for the widths of the layers of a deep network to scale polynomially with the number of training data. For example, the CIFAR-10 training dataset contains 50000 images. Thus, if the layer widths were to scale *only linearly* with the number of data samples, the total number of parameters would be roughly $2.5 \cdot 10^8$. This is way more than what it suffices to completely fit random labels on CIFAR-10, namely $N \approx 10^6$ [72]. ImageNet is even bigger: even if we consider a subset of $1.2 \cdot 10^6$ images as in [72] (out of the total $1.4\cdot 10^7$ training data samples), a similar back-of-the-envelope computation leads to a model with more than $10^{12}$ parameters, which surpasses in size the largest model ever released. Again, in [72], it is shown that $2.4 \cdot 10^7$ parameters are enough to fit random labels for this subset of $1.2 \cdot 10^6$ training data points.
>
> The numerical evidence presented above suggests that having a number of parameters of the same order as the dataset size is much closer to practice than having a number of neurons of that order. The main result of our paper gives a **rigorous justification to this phenomenon**, showing that memorization and optimization are in fact possible as soon as the number of parameters (and *not* the number of neurons) scales at least linearly in the dataset size. Such a result can be regarded as the end point of a popular line of research in the theoretical deep learning literature, which has provided optimization guarantees under milder and milder over-parameterization requirements. More specifically, [5] requires the layer widths to be $\Omega(N^{24})$, [73] requires $\Omega (N^{14})$ neurons, [74] requires $\Omega (N^{8})$ neurons, [21] requires $\Omega (N^{4})$ neurons, and finally [47] requires $\Omega (N)$ neurons. Furthermore, there is increasing evidence that, in the challenging setup in which the layer widths are *sub-linear* in $N$, neural networks still memorize the training data [14, 71, 66], reach zero loss under gradient descent training in the two-layer setting [61, 50, 42], and in the deep case, gradient descent explores a nicely behaved region of the loss landscape [44]. We also remark that in some of the existing work (e.g., [44]), it is explicitly left as an open problem to prove that gradient descent is able to optimize neural networks with sublinear layers.
>
> In summary, **our work addresses this open problem, and it is the first to show that a class of deep networks with minimum over-parameterization can memorize and optimize**. To achieve this goal, we need to develop a novel analysis, as also acknowledged by *Rev. mnfH*, who recognized our technical innovations.
>
> In order to address this point, we have added a clarification in l. 36-40 of the revised version.
>
> ----
>
> **Question 2**
>
> In the body of the paper, we mention several times that the NTK is shown to be “well conditioned”. When we do that, we refer to the conditioning number of the matrix, given by the ratio between its largest and smallest eigenvalues. Since we prove a lower bound on the smallest eigenvalue of the NTK, this automatically gives an upper bound on its condition number, thus showing that the NTK is in fact well conditioned.
>
> We hope that this clarifies the concern of the reviewer, and we remain at disposal in case further clarifications are needed.
>
> ----
>
> **Question 3**
>
> This is an excellent point. As correctly guessed by the reviewer, the labels used in Figure 2 are in fact random. However, similar results can be obtained using ‘low-complexity’ labels. In particular, we have repeated the experiment using as labels $y_i = v^\top x_i$, where $v$ is a random unitary vector in $\mathbb R^d$. The results are very similar and they are provided in the file `optimization_labels.pdf` in the supplementary material of the revised version.

---

### Author Response · Authors · 2022-08-01
**Overall response to the reviewers**

We would like to thank the reviewers for their numerous valuable comments. We are glad that the reviewers recognize that our “proofs are solid” (*Rev. 5UpM*) and “the theoretical contributions are clear” (*Rev. mnfH*). We are also thankful to *Rev. mnfH* for acknowledging the technical innovations of our argument.

Next, we provide separate replies to the four reviewers, in which we address all their points. We have uploaded a revision of the body of the paper and of the supplementary material as well. In our responses, we point to the parts of the paper that have been modified. The numbering of lines, equations and references refers to the revised version, and the main changes are highlighted in blue color. We have tried to keep such changes to a minimum in the body, in consideration of the strict 9-page limit holding also for the revision.

Finally, we remark that we have been able to remove the following assumption on the activation function: $\mathbb E_{\rho} [ \phi(\gamma \rho) ] \neq 0$ for all $\gamma \neq 0$, where $\rho$ is a standard Gaussian random variable. This was the third part of Assumption 2.3 in the submitted version and we had already pointed out that this “last condition is purely technical and simplifies the part of the argument concerning the centering of the feature maps”. In fact, after the submission deadline, we have found a bug in the proofs of Appendix D and, after writing down a simple way to fix the bug, it became apparent to us that the assumption above was unnecessary. As a result, we have removed the extra assumption, updated the discussion on the centering part (see l. 223 and l. 233 of the revision) and also the proofs in Appendix D.

---

### Meta-Review · Area_Chair_xJrp · 2022-08-24

**Recommendation:** Accept
**Confidence:** Certain

**Metareview:**

solid contribution to ntk theory

**Award:**

No

---

### Decision · Program_Chairs · 2022-09-14

Accept